# Possibilities of Using UAVs in Pre-Hospital Security for Medical Emergencies

**DOI:** 10.3390/ijerph191710754

**Published:** 2022-08-29

**Authors:** Marlena Robakowska, Daniel Ślęzak, Przemysław Żuratyński, Anna Tyrańska-Fobke, Piotr Robakowski, Paweł Prędkiewicz, Katarzyna Zorena

**Affiliations:** 1Department of Public Health & Social Medicine, Medical University of Gdańsk, 80-210 Gdansk, Poland; 2Division of Medical Rescue, Faculty of Health Sciences with the Institute of Maritime and Tropical Medicine, Medical University of Gdańsk, 80-210 Gdansk, Poland; 3Department of Anesthesiology and Intensive Care, Oncology Center—Memorial Hospital in Bydgoszcz, 85-796 Bydgoszcz, Poland; 4Department of Finance, Faculty of Economics and Finance, Wrocław University of Economics, 53-345 Wroclaw, Poland; 5Department of Immunobiology and Environmental Microbiology, Medical University of Gdansk, 80-211 Gdansk, Poland

**Keywords:** drones, medical security, disaster management, emergency medical services, security, costs

## Abstract

The term unmanned aerial vehicle (UAV) was post-applied in the 1980s to describe remotely piloted multi-purpose, unmanned, autonomous aircraft. The terms unmanned aircraft systems with data terminal connectivity (UAS) and remotely piloted aircraft systems (RPV, RPAS—military systems) are also used. This review aims to analyze the feasibility of using UAVs to support emergency medical systems in the supply and urgent care ranges. The implementation of drones in the medical security system requires proper planning of service cooperation, division of the area into sectors, assessment of potential risks and opportunities, and legal framework for the application. A systematic literature search was conducted to assess the applicability based on published scientific papers on possible medical drone applications in the field of urgent mode. The widespread applications of UAVs in healthcare are concerned with logistics, scope, and transportability, with framework legal constraints to effectively exploit opportunities for improving population health, particularly for costly critical situations.

## 1. Introduction

The term unmanned aerial vehicle (UAV) was post-applied in the 1980s to describe remotely piloted multirole, unmanned, autonomous aircraft [1]. The terms unmanned aircraft systems with data terminal connectivity (UAS) and remotely piloted aircraft systems (RPV, RPAS—military systems) are also used [2]. This article will use the terms UAV and drone as the broadest and most commonly used terms in civilian use, including medical use. With the development of technology, unmanned aerial vehicles (UAVs) can find applications in many sectors of the economy and human life. Current UAVs have become very efficient, accessible, and affordable. The categorization of drones is related to their size, performance, and legal requirements in this regard [3]. They are increasingly being used in the management of urgent mode situations as standard equipment for managing emergencies such as epidemics, mass accidents, or rescue missions in hard-to-reach areas. Since it is possible to use these technologies directionally in warfare by eliminating human targets [4], given the efficiency of the detection and tracking concept based on acousto-optic technology and active imaging [5], these technologies can also be used for pro-social emergencies.

UAVs have already been applied to medicine in many European countries, and research into the possibilities of better applications is ongoing. One of the first programs to use UAVs is the UNOSAT program and UNITAR [6]. The goals are, among others, humanitarian human security, strategic urban planning, and strategic development planning. Drones are thus being used as an assessment tool for natural disasters and as a means of supporting emergency services [7]. In July 2015, for example, they were used to deliver life jackets to individuals stranded on rocks in Maine [8]. Drones in emergencies have thus proven themselves for many years, reducing the response time of emergency services [9] also in Poland [10].

## 2. Purpose

This review aims to analyze the applicability of UAVs to support emergency medical systems in the ranges of delivery and emergency urgent care.

The implementation of drones in the medical security system requires proper planning of service cooperation, division of the area into sectors, assessment of potential risks and opportunities, and a legal framework for the application.

## 3. Methodology

A systematic literature search was conducted to assess the feasibility based on published scientific papers on possible emergency medical drone applications. The Web of Science service was used as a search engine. An advanced search was conducted to identify sources containing the phrases “drone”, “UAV”, and “unmanned aerial vehicles” as subject terms. 

The sources were arranged chronologically, and their titles were checked for relevance and selected where deemed appropriate. No document type was excluded. The scoping review method was also used

The search was not limited to the medical field. Web of Science categories were searched from the fields of operation research management science, public environmental occupational health, economics, ergonomics, emergency medicine, business, healthcare science services, hospitality leisure sports tourism, medicine legal, surgery, behavioral science, agricultural economy policy, business finance, medicine research experimental, public administration, psychiatry health policy services, pediatrics, infectious diseases, virology, multidisciplinary science, biochemical research methods, medical informatics, and management.

Sources were selected that discussed the use of drones of the following industries: 427 Operations Research Management Science, 251 Engineering Electrical Electronic, 282 Multidisciplinary Sciences 203 Automation Control Systems, 93 Computer Science Artificial Intelligence, 89 Applied Mathematics,85 Computer Science Information Systems, 75 Biochemical Research Methods, 60 Engineering Industrial, and 60 Telecommunications.

Types of sources included journals, scientific journals, news articles, trade publications, and electronic resources. All sources published in English in 2017-VII 2022 were included—due to the rapid development of the drone market and changing laws. Duplicate search results were excluded. Academic sources were defined as those published in scientific journals and conference proceedings. Relevant literature was extracted from these articles. The search results showed 960 publications from the following publishers: 250 IEEE, 122 Elsevier, 110 Springer Nature, 57 MDPI, 53 Public Library Science, 39 Humana Press Inc. 38 Systems Engineering and Electronics, Editorial Dept, 35 Taylor and Francis, 34 Wiley-Hindawi, 20 NATURE PORTFOLIO. The detailed process of conducting a literature analysis is presented in Figure 1.

The work is divided into 3 categories, from the determination of optimal paths to reach the threat trajectory, networks to regulate, and restrictions on use through the main part of the analysis on mobility and type of transportation in an emergency mode situation.

## 4. Results

The purpose of the analysis concerns the scope and applicability of drones to support emergency medical systems in the ranges of transport and emergency urgent care. Modern public services can conduct complex operations in times of emergencies of any kind. The tasks of firefighters or the ability of paramedics to use equipment/medicine have expanded. Guidelines from INSARAG [11], an intergovernmental organization implementing the International Strategy for Disaster Reduction (ISDR) [12], have emerged. The MOBNET system was created to create a tailor-made solution to support the traditional activities of search and rescue groups supported by DCT-EGNSS technology for casualty tracking [13]. In 2018, the Civil and Military Air Rescue Coordination Center (ARCC) was also launched at the Polish Air Navigation Agency (PAŻP), for the coordination of search and rescue activities in the ASAR (Aeronautical Search and Rescue) service in the Polish area of responsibility [14]. In the world, UAV search and rescue systems are proposed [15] as distributed and modular coordination frameworks for swarms of UAVs oriented to mission-oriented target recognition and tracking [16] or effectively in collision-free navigation of UAV fleets [17]; however, only the combination of all drone activity markets can bring tangible benefits to the emergency medical care mode. Currently, in Poland, the number of drones in the public space is estimated to reach 100,000, with 90 training centers and 10tys of people holding qualification certificates for commercial flights (Q1 2019) [18].

## 5. Possibility for Determining Optimal Paths to Reach a Threat-Trajectory, Networks

There are several UAV search studies aimed at minimizing search time and maximizing the probability of detection or human search support by the drone [19]. Search algorithms are time-consuming and sometimes not very practical. A new approach is an evolved algorithm using biogeography-inspired drone operators who have been able to reduce the time it takes to reach a target by about 79 and 147 min, providing significant improvements in life-critical operations [20].

Drones now also have live tracking capabilities. Drones prevent drowning by providing first aid while keeping rescuers away from dangerous conditions at sea [21]. Drones track fires, which can hinder operations requiring immediate life-saving action, and measure the moisture content of soils. However, the biggest challenge in this area is routing drones. One model is routing using simulation-based optimization approaches to mitigate fire risk based on algorithms. A multi-objective evolutionary algorithm searches the space of potential field parameters to maximize fire coverage while minimizing energy consumption. Fire spread is modeled [22]. Mathematical models are used to generate UAV routes used to detect fires, such as forest fires. The optimization algorithm is based on simulation using real-life dynamics with the application of fire probabilities when calculating the suitability/target values of the routes [23]. Models are also being extended by integrating more uncertainty factors (e.g., wind, temperature, humidity, etc.) with their real-time values. Soil moisture is already being estimated by analyzing UAV hyperspectral images with optimal metrics and a set of field observations within a machine learning framework will provide highly accurate SMC estimation [24]. There is also an opportunity to use a low-cost hybrid ground-to-air sensor network to measure environmental parameters using optimal trajectories. UAVs are used to measure environmental parameters directly, as well as to collect data from ground sensors while improving energy consumption and data acquisition efficiency [25].

Computationally efficient route planning to avoid UAV collisions is being developed [26]. Different route trajectories, to generate the optimal route the so-called orienteering problem of UAV routes [27], smooth trajectories [28], or action path scheme planning [29], most optimal choices are based on mathematical modeling and algorithms. Dynamic programs with a hierarchical directional algorithm can improve terrain usability while greatly reducing computational complexity [30]. Of note is effective path planning for an unmanned aerial vehicle (UAV) that tracks a narrow area target in a complex environment [31]. Secure continuous connectivity of the UAV’s communication and control system is also used, even during the occurrence of a network switch [32]. A July 2021 literature review on routing through drone use lists heuristic algorithms and meta-heuristic methods along with action policies possible at the current time for use in optimizing the time, distance, and cost of drone operations. Several logistics applications for shipment distribution and information gathering were described, as well as attention was given to combining drones with ground vehicles. The battery/fuel issue was identified as the biggest limitation [33].

Thus, drones are used in urgent day-to-day operations, but it is useful to plan the entire system. An example is Manhattan, where the planning of roadside units (RSUs) is crucial to the operation of an intelligent transportation system (ITS) that can be used in emergencies [34]. The cooperative search method for multiple UAVs for a moving target is more efficient. Cooperative search strategies for multiple UAVs with a V-formation shape increase the probability of detecting a moving target by reducing the blind area of airborne sensor coverage, which will increase the probability of detecting a moving target [35].

The use of UAVs in logistics distribution, especially in complex terrain, both urban and remote from human concentrations, can be used together with a ground vehicle. One needs communication, route planning, and optimization of time and energy consumption in delivery. The results show that the UAV and vehicle can work together to complete delivery of all customer demand nodes, and joint delivery of the UAV and vehicle can effectively reduce the total delivery distance. Once laws and regulations related to UAVs are improved, flights and thus deliveries can be expanded to more urban areas for logistics distribution [36]. There is also autonomous cooperation between ground robots and aerial robots, with coordination and coordinated object transportation control schemes through the heterogeneous formation of UGV and UAV [37]. However, there is a problem with routing and route planning for trucks cooperating with multiple drones, drone truck tandems are mainly limited to UAV launch and recovery operations (LAROs) to customer locations, a new variant of the drone truck tandem has been introduced that allows the truck to stop at locations other than the customer, showing a significant improvement in delivery performance by using flexible locations for LAROs, as opposed to the existing approach of limiting truck stop locations [38]. This solution is already directing the possibility of optimizing a transportation system that works together with a truck and drone to deliver medical resources in a broad sense. 

Since, at the same time, business opportunities for drone services are growing with the automation of services, even without the human factor, that makes services cheaper and faster [39], we can also, by focusing on transportation and delivery, come to the simple conclusion that in a disaster situation these activities will also be faster and cheaper. However, since during a disaster there is usually destruction of the local communications infrastructure (buildings, power sources, and transmitters), drones as flying base stations can be a solution to restore basic communications services in emergencies. Optimization of a real-time resource allocation scheme for UAV-assisted relay systems in emergencies, such as rescue and public safety missions, has been implemented [40]. A mini-review of the mobility model of flying ad hoc networks in disaster areas shows that drones are becoming useful in emergencies and hard-to-reach areas, especially during disasters. An effective mobility model for reconnaissance and aid scale-building, as mobile technology has reached high throughput [41]. In this regard, one can also see the development of unmanned aerial vehicle (UAV) swarm research for better situational awareness. The primary goal of UAV swarming is to gather as much information as possible. New algorithms for patrol task scheduling for UAV swarming improve patrolling efficiency, guaranteeing performance [42]. Another search scheme is using the elite learning method for multiple bee colonies for UAVs in unfamiliar environments. In this research, a cooperative search for multiple dynamic targets in an unfamiliar environment by a so-called swarm was applied. Based on the elite learning algorithm for multiple bee colonies (MBC), it improves the adaptability and computation speed of the standard artificial bee colony (ABC) algorithm in various search missions [43].

Unmanned aerial vehicle (UAV) applications, therefore, make rescue operations as well as everyday life easier. However, they can also carry third-party risk issues. A UAV can fall and collide with pedestrians or vehicles, it can lose controllability and destroy property or damage assets causing property damage, not to mention the problem of noise. Correct route planning is a method of mitigating these risks and their effects. Unfortunately, most existing route planning methods focus on minimizing flight distance or energy costs, without considering the costs associated with risk. One paper describes a novel flight path optimization method that also considers an integrated cost assessment model. The evaluation model includes the risk of death, property damage, and the impact of noise in the city of Singapore. Simulation results showed the effectiveness of the cost-based path optimization model in reducing risk costs—a decrease in costs by 42.64% and 44.15% at a 95% confidence level [44]. The concept of Unmanned Traffic Management (UTM), adopted in Europe as U-Space in the Warsaw Declaration, has initiated several projects enabling drone flights at low altitudes, but for wider use, a functional and optimal U-Space architecture for the Single European Sky (SES) needs to be designed [45].

A limitation of drone use is the power system. UAVs often waste a lot of time and fuel to take off and reach difficult places, places covered by warfare, difficult terrains including cities, etc. [46]. To prevent this, two design concepts for rapid deployment of UAVs have been developed: the first is a ground-based Rapid Deployment unmanned aerial vehicle (RDUAV) for vertical takeoff, and the second is an airborne rapid deployment unmanned aerial vehicle (AL-RDUAV) to reach the target quickly eliminating the time gap and leaving the full advantages of UAVs. Solutions of this type can respond quickly during wartime missions or other emergencies [47]. Keeping in mind the problems of infrastructure in terms of emergencies, unmanned aerial vehicles (UAVs) can be used to support transportation communications by acting as a communication relay [48]. Keeping in mind the effective management of energy while collecting information in the field, various systems have been proposed as a minimum a system consisting of RFID tags deployed in the field and a reader installed on the unmanned aerial vehicle (UAV). The UAV is used to collect data from RFID sensors scattered throughout the area by simply approaching them, flying over them, and collecting the measured data. This solution can be used to deploy a grid of independent RFID sensors covering a large area or to query sensors located in situations that are dangerous to people [49]. At the same time, drones equipped with a high-resolution digital camera are being used to visually assess the health of a building enabling strategic decisions on the maintenance and sustainability of building assets, being able to use these activities in medical support systems [50]. There also follows the use of UAVs for technical inspections in a population-based predictive approach as an interesting data collection tool [51]. All of these activities can be useful in a health emergency mode.

The problem of optimizing energy intake/storage has shifted to solar-powered drones. Compared to a traditional drone, a solar-powered UAV (SUAV) drone has longer endurance, converting solar energy into electricity keeping in mind the effect of different solar irradiance on optimizing the SUAV’s trajectory [52]. The energy management scheme is also used in landing an unpowered drone. Once the drone completes its cruise, it shuts off the engine and begins to land, reducing weight [53] and lowering the energy cost by up to 70% [54,55,56]. Drones also support forensics, increasing detection accuracy compared to traditional human approaches. In particular, a drone can provide detection rates of nearly 100%. Increased detection speed over large areas, and high detection rates in different types of terrain—thanks to computer vision techniques [57,58,59]. To save lives during emergencies, it is also possible to detect the human body using UAV camera image processing. Both the human body and any part of it can be detected [60]. At the same time, since UAV bioacoustic monitoring has also been introduced, it is also possible to block propeller noise from the receiving microphone and report the characteristics of bioacoustic recordings. In the case of bats, since there are no other studies in this area, it is a very promising method [61].

## 6. Mobility and Mode of Transportation

The epidemic has forced the healthcare system into a non-contact mode of communication and delivery of goods, and the war has forced rapid response and precision delivery. Medicines, blood, and AEDs are essential for saving lives, treating diseases, and responding to emergencies, with the likelihood of disruptions in standard supply chains. Using drones to distribute medical aid, information was obtained from the range of the required number of drones and the locations of the aid centers of the cases studied. 

Drones are used communicatively for diagnosis, treatment, perioperative assessment, and the support for those on the scene, so it places their use on national and international agendas [62]. Keeping in mind the above conclusions, and the limitation in travel distance there is a need for charging stations or alternative methods to extend coverage, finding the best topology with minimal cost and waiting times. A study from Istanbul on optimizing the transportation of blood products from distribution centers to hospitals in cities using UAV routes proposed an optimal path and type of UAV while minimizing the number of UAVs used, optimizing the route, range, weight, and volume of UAV payload. Test results show that the proposed methods can find good solutions to the problem in acceptable CPU times [63]. However, the emergency poses greater risks in medical transportation. Inaccessible infrastructure may require sudden coordinated system actions. In a review of the current state of innovative drone delivery and decision-making models in this area, with a focus on healthcare, it was shown that retrospective analysis generates innovation and two new models related to the drone healthcare network design have been created facilitating cost-effective, timely, and efficient drone healthcare delivery service [64].

A 2019 study proposed a hybrid algorithm to solve the model, providing increased efficiency and flexibility in the disaster relief system [65]. With this in mind, a multipurpose algorithm for delivering blood via drones to the injured in an emergency was created. As a complex problem including the limitations of blood transport and minimum maintenance of the temperature of the delivered blood during transport, attention should be paid to the typical problems of the drones described above, i.e., scheduling and routing, and limited carrying capacity. The described algorithm has achieved better optimization and time efficiency results and, in addition to blood, can be used for heating food and scheduling food delivery routes, transporting frozen seafood, etc., via UAV [66]. The integration of UAVs with intelligent transportation systems (ITS) may strengthen services for smart cities. The logistical efficiency and network communication of the hybrid delivery fleet are optimal [67]. The use of UAVs in commerce is restricted by law (Federal Aviation Administration (FAA), and the transport of critical medical equipment is often limited to wheeled ground vehicles and manned aircraft. In 2015, it was predicted that delivery of UAV systems would be a financially, legally, and technically feasible means of transporting medical products [68]. This is the case today. Any analysis of integrating drones into the healthcare supply chain involves decisions related to the selection of UAVs for medical supplies in a broad sense. There are many types of drones with different characteristics. Characteristics include flight range, payload, or battery power/type, and making the optimal decision as to which drone is needed for particular transport is a challenge for decision makers. Accordingly, a decision support model was developed to select the optimal type of drone, for now for two specific scenarios related to the delivery of medical supplies, using a methodology that incorporates graph theory and matrix approach (GTMA). It turned out that drones equipped with cargo handling capability and parcel handling flexibility are more preferred in scenarios with urban areas, while drones with longer flight distances are most often prioritized in disaster scenarios where the road communication system is destroyed or inaccessible. This analysis covered so-called critical gaps in the existing literature by formulating a mathematical model to find the most suitable drone for a specific scenario based on its criteria [69]. However, it is not just about UAVs. In this regard, the cooperation of the vehicle transport system with the UAV is an underestimated help. Such logistics systems can deliver equipment, emergency supplies, medicines, etc. To any place on the world map. The number of trucks has a greater impact on the optimal solution than the number of drones, and the performance of UAV swarm optimization is better than other algorithms. However, in a cooperative transportation system between a truck and a drone, optimization of route planning is a very complex problem [70]. When delivering goods that require very restrictive transportation methods (time, temperature), such as blood products, it is also worth considering the choice of drone or ground transportation, or perhaps assisted ground transportation. The average UAV transport time was significantly faster than ground delivery (17:06 ± 00:04 min vs. 28:54 ± 01:12 min, *p* < 0.0001), and the mean ± SD initial temperature for packed red blood cells were 4.4 °C ± 0.1 °C with a maximum variation in mean temperature of 5% from takeoff to landing. Unmanned air transport of simulated blood products was much faster than ground transport. The temperature of simulated blood products during transport remained within appropriate acceptable ranges. Further studies evaluating UAV transport of real blood products in populated areas are warranted [71].

Electric defibrillators are marginally easier to transport than blood, for example, although topography and weather still have a major impact on drone delivery of AEDs for out-of-hospital cardiac arrest. Since out-of-hospital cardiac arrest (OHCA) is one of the leading causes of death worldwide, affecting an estimated 275,000 people in Europe each year [72], early defibrillation is one important element in improving survival rates for OHCA patients [73]. Those seeking to provide community-based first aid while improving survival rates in out-of-hospital cardiac arrest (OHCA) face obstacles in the form of limited public access to defibrillation. Whether this is due to a lack of equipment or ignorance about where it is installed. The deployment of new technologies over time reduces the cost of their use, so there are increasing attempts to use UAVs in emergencies, including humanitarian crises at different levels of application, which are mapping, reconnaissance, logistics, and transportation. UAVs can deliver automated external defibrillators (AEDs) directly to the OHCA site or support the use of the equipment by bystanders. Several public-access defibrillation programs have been implemented in various countries [74,75]. It is difficult to find the nearest PAD and apply it quickly, thus overcoming fear [76]. For these reasons, UAV delivery of automatic electrical defibrillator (AED) has begun to be investigated, which can reduce the time of AED application and the time of defibrillation itself [77,78]. Previous research in this area would describe several limitations, from routing with all its topography and meteorological constraints to the regulatory framework in a country [79,80]. The use of UAVs to transport AEDs has the effect of reducing AED waiting times in metropolitan cities, but not in bad weather [81,82]. In rural areas, on the other hand, a 2022 study found that drones delivering automated external defibrillators, unfortunately, remain an experimental technology, although integrating them into the survival chain is feasible and safe, and potentially improves the availability of early public defibrillation, especially in rural areas [83]. At the national level, geographic information system (GIS) analyses can identify areas of high OHCA prevalence and serve as tools to quantify the need for AED-equipped drones; a few drones can significantly increase OHCA protection coverage [84]. Experimental technology has developed a simulation framework for assessing humanitarian logistics during states of disaster, war, and epidemics—logistics that often include humanitarian assistance in a wide range that can be quickly adapted to a specific disaster scenario [85]. Drones are also proving their worth in protecting the health and lives of humanitarian aid workers. The idea of the so-called humanitarian flying warehouses (HFW), or high-altitude airships using UAVs to deliver supplies, has emerged. Such an idea optimizes interruptions in the supply path in both conflict zones and emergencies [86].

Delivery of an automated external defibrillator via UAV has potential, and it was found that retrieving an AED delivered by a drone was perceived as safe and feasible for bystanders, even accompanied by a sense of relief upon arrival of the drone with the AED and interaction with the drone AED was perceived as less difficult than performing CPR or using one’s cell phone during CPR [86,87]. However, not every community is ready for such innovations. In some communities, there is a need for additional investment to close the pre-hospital response gap (min improved emergency management structures) and support local communities as first responders in local emergencies [88]. Integration of technology, however, should be coupled with support systems, investments, and respect for cultural values [89]. An important aspect of the whole system of UAV use is the public’s attitude toward drones as a social aspect of security improvement [90]. The compatibility of drone delivery with consumers’ lifestyles, environmental friendliness, reliability, safety, or novelty should be considered [91]. Additionally, the increasing use of UAVs for terrorist activities does not help the process of expanding UAV systems [92].

Drone applications in mass disaster management are rarely described. An analysis of current projects that were created in 2022 evaluating current drone use projects in mass disasters found that there is currently insufficient evidence to warrant a systematic review of drone use during disasters [93]. The use of UAVs in humanitarian assistance to search for victims can be effective, including in terms of cost. However, the issue of privacy and access to airspace is limited and needs to be regulated [94]. The use of drones for medical security at mass events is also related to the cost of purchasing and maintaining UAVs, and issues of training personnel and drone operators [95].

In emergency rescue, UAV support is seen in three aspects, real-time monitoring, search and rescue of disasters, and post-event assessment [96]. UAV search and rescue operations, combined with a real-time machine learning-based object detection system, achieve 94.73% accuracy and can maximize the chances of saving lives—a system deployed by Scotland police to help SAR teams locate missing persons [97]. The architecture of many drones with cellular communication for disaster management by predicting the movement scenarios of human gatherings (gatherings, stadiums, etc.) also adapts sensory models for disaster monitoring and achieves 99% accuracy in disaster detection [98]. It can also provide information to emergency services both by generating immediate crisis maps for affected areas and human concentrations, providing an important tool for promptly prioritizing/managing SAROs efficiently and effectively [99].

At the same time, it is worth noting the improvement of scene size as a key factor in the first stage, mass casualty incident (MCI) response. UAVs have the potential to increase scene size through visual feedback in real-time during sudden, changing events. A study of paramedics’ actions using UAV video from a simulated mass casualty incident in segregating patients and creating key operational locations was conducted. It turned out that remote scene creation in mass casualties was correct. The majority of participants correctly sorted and correctly located patients, and selected an acceptable location to begin stage two of SALT selection by identifying hazards and determined key operational areas [100]. Using the UAV as a tool for assessing emergency medical situations showed no statistical difference in most variables between the UAV and the real situation assessment [101]. Thus, despite the experimental view of UAVs in support of the rescue system, it can be thought that within a few years it will be possible to support operations with remote systems. There are also rescue applications that benefit from mapping through the use of UAVs [102].

## 7. Regulations and Application Restrictions

In June 2019, the EU Commission’s regulations numbered 2019/947 replacing 2020/746, 2019/945, 2020/639, and 2020/1058 were published in the Official Journal of the European Union regarding rules for drone use in the EU [103]. As of 2020, 340 drones and more than 15,500 qualification certificates have been registered in Poland, and more than 110 companies are involved in unmanned flying device training activities [104]. The current three categories of flight: open special and classified are fraught with restrictions. Open without ULC govt, low risk, up to 25 kg, maximum altitude 120 m; special information to ULC, medium risk, flight conditions related to risk analysis; certified—flight certification related to personnel and equipment due to high risk.

According to the Civil Aviation Authority, operations in the open category allow low-risk flights and do not require a prior flight permit from the CAA [105]. They can only be carried out within the visual range of the pilot or observer, no further than 120 m above the ground, with drones weighing a maximum of 25 kg. The category is divided into subcategories A1, A2, and A3 according to the requirements for pilots of unmanned aerial vehicles and drones. Where A1 allows the flying over bystanders but not over gatherings of people, A2 is a ban flying over people and gatherings of people, with a minimum horizontal distance from people of 30 m or 5 m if the drone has a speed-limiting function, and A3 is a ban on flying over people and gatherings and at a safe distance of at least 150 m horizontally from residential, commercial, industrial, or recreational areas. Notably, certificates of competency for unmanned aircraft pilots and permits for operators of unmanned aerial systems are mandatory [106].

Unfortunately, regulations both promote and stifle innovation. Regulations and systems in many countries around the world are not yet mature enough to regulate modern support methods such as ground transportation. All stakeholders including regulators, users, and manufacturers must work together to optimally and effectively regulate the new technology [107]. Unfortunately, most research on UAV safety focuses on building algorithms and transmitting information, and possibly basic hardware safety. There is an apparent information gap on the issue of security in the broadest sense because UAV security research is linked to modeling, algorithms, information, computation, transmission, network building, operation, etc. The public is aware of UAV safety issues, but due to emerging technologies, this knowledge is not optimal, as safety research has not kept up with technological developments. We still need regulation, oversight, and also price coordination in UAV development [108]. There is, of course, emerging research related to at least calculating the risk of ground accidents during UAV flights, where recommendations are being made to improve safety when designing flight paths for drones in built-up areas [109]. Finally, it is worth noting the mental workload of UAV controllers. The difficulty of the mission significantly affects the burden of professional burnout and the number of failures, as the heavy mental workload of the UAV can lead to poor flight performance [110]. Despite the above, the behavioral intention to use drones in rescue missions is due to the expected increase in performance and favorable conditions. Rescue organizations are advised to consider conditions in implementing drones in rescue logistics, and drone manufacturers are advised to focus on operational performance while providing support and training to users. The importance of personal and environmental factors in the acceptance of drones in emergency logistics emerges as key here [111]. The use of drones in the broader healthcare field is viewed positively by medical professional groups. The driving factors here appear to be an innovative environment, experience with cutting-edge technologies, and support for new ideas [112].

## 8. Conclusions

The increasing proliferation of drones’ affordability in recent years in private use has prompted proactive measures to prevent disasters and mitigate emergency mode events [113]. Drones are increasingly being used to support healthcare in the broadest sense. However, to properly and optimally use these technologies, special attention must be paid to regulation and safety, selection of appropriate technologies and methods of operation (routing), and coordination of market stakeholders for unbiased creation. To accelerate the use of drones in healthcare, there is a need to share experiences and evidence from ongoing projects [114].

The widespread use of UAVs in healthcare is concerned with logistics, scope, and transportability, with framework legal constraints to effectively use the capabilities to improve the health of the population, especially in costly critical situations. To use drones to save lives around the world, the limitations of the system still need to be met [115]. At the same time, additive UAVs used to prevent unwanted activities in human concentrations, and preventing emergencies through surveillance of human attitudes are needed [116]. Thus, the potential applications of UAVs in medicine are wide-ranging. UAVs deliver vaccines, automated external defibrillators, medicines, and blood products, identify victims and assist in forensics, secure remote areas difficult to supervise, and safeguard public health. The use of drones in medicine can both increase the quality and accessibility and lower the cost of healthcare itself [117]. Moreover, their use on an urgent basis will further speed up the time to act, which is crucial in a crisis.

## Figures and Tables

**Figure 1 ijerph-19-10754-f001:**
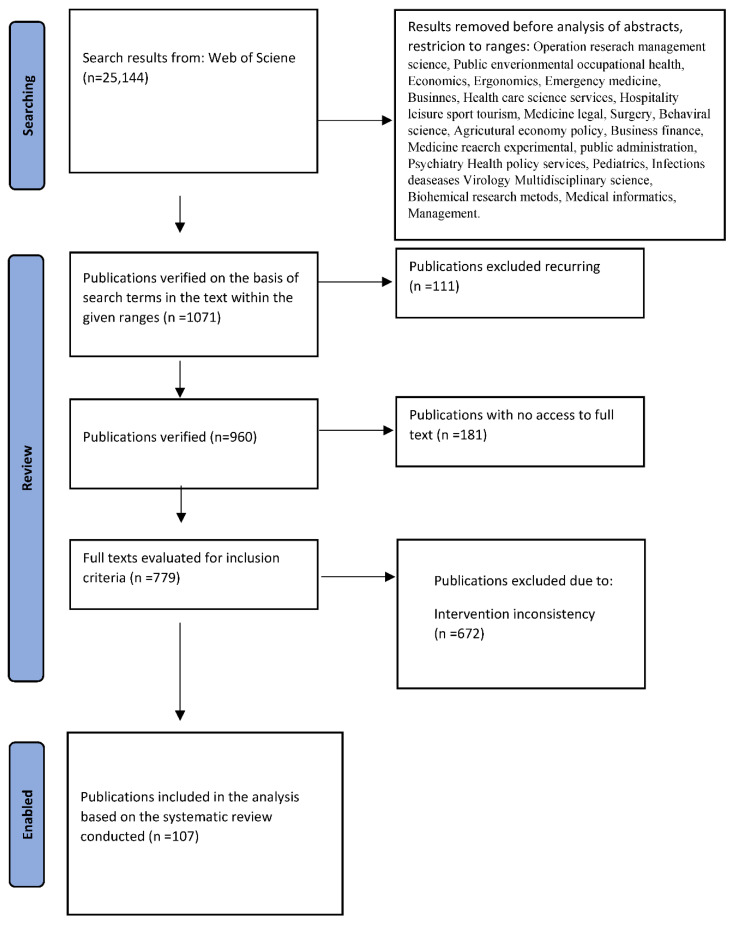
Literature analysis.

## Data Availability

Data sharing not applicable.

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
