# Peer review of "Possibilities of Using UAVs in Pre-Hospital Security for Medical Emergencies"

_ijerph, 2022, doi:10.3390/ijerph191710754_

Round 1

Reviewer 1 Report

The peer-reviewed manuscript discusses an essential element of emergency medical support through the use of an unmanned aerial vehicle (UAV). From the point of view of the development of technologies and forms of providing emergency medical services, any mechanism supporting this system is critical. Hence, the topic taken up by the authors is both scientifically and clinically relevant.

The review's methodology is not objectionable. The authors searched the Web of Science database using the correct keywords (a correction probably needs to be made to line 66/67 where the authors indicated the 'Web of Scince' database). 

In a PRISMA-compliant flow diagram, the authors presented the decision-making process for including individual articles in the review. A total of 107 publications were ultimately reviewed. Such source material ensures a good level of content for the work. Regarding the flow diagram presented, the figure does not contain a legend, caption, or reference in the text. In the version submitted for review, the blocks on the left are of poor quality (white bars on the figure) - this does not detract from the merits of the paper but is worth correcting before eventual publication.

In discussing the results, the authors divided the results into two parts: the use of the UAV as a means of optimising the search for reaching a destination, etc., and in the second part, they discussed the use of the UAV as a means of transport. The authors also discussed legal regulations and various aspects of restrictions on UAV use. The results were discussed at great length, referring to examples - also pointing out the risks of UAV use. Each of the above sections has been discussed extensively and critically.

Substantively the work is of a good standard. Editorial aspects need to be standardised, e.g. the way footnotes are placed - in parts of the text, the authors place square brackets before the full stop at the end of the sentence and in some places after the full stop. 

Author Response

Thank you for your favorable review and the suggestions you received. We improved the readability of the figure and added information about it in the manuscript. We also improved punctuation and references.

Reviewer 2 Report

This article is very interesting; highlights the Possibilities of using UAVs in pre-hospital security of medical emergencies. I feel it can be accepted as it stands.

1) A systematic literature search was conducted to assess the feasibility based on published scientific papers on the possible applications of medical rescue drones. The Web of Scince service was used as the search engine. An advanced search was conducted to identify sources containing the phrases "drone," "UAV" and "unmanned aerial vehi-cles" as subject terms. The sources were arranged chronologically, and their titles were checked for relevancy and selected where deemed appropriate. No document type was excluded. A scoping review method was also used.

The search was not limited to the medical field. Web of Science categories from the fields of Operation research management science, Public environmental occupational health, Economics, Ergonomics, Emergency medicine, Business, Health care sci-ence services, Hospitality leisure sports tourism, Medicine legal, Surgery, behavioral science, Agricultural economy policy, Business finance, Medicine research experi-mental, public administration, Psychiatry Health policy services, Pediatrics, Infections diseases Virology Multidisciplinary science, Biochemical research methods, Medical in-formatics, Management.

2. Very little work in the field on the use of drones in emergency medical care.

3. The meta-analysis compares the use of drones in every aspect

4. For the future, a limitation in the number of articles to 100, due to the long readership

5. the conclusions are consistent with the evidence and arguments presented and they address the main question posed.

6. the references are appropriate

7. Figures are present in the text could be clearer

Author Response

Thank you for your favorable review and the suggestions you received. We improved the readability of the figure.